# Serum C18:1-Cer as a Potential Biomarker for Early Detection of Gestational Diabetes

**DOI:** 10.3390/jcm11020384

**Published:** 2022-01-13

**Authors:** Ilona Juchnicka, Mariusz Kuźmicki, Piotr Zabielski, Adam Krętowski, Agnieszka Błachnio-Zabielska, Jacek Szamatowicz

**Affiliations:** 1Department of Gynecology and Gynecological Oncology, Medical University of Bialystok, 15-276 Bialystok, Poland; ilona.sikora@ubm.edu.pl (I.J.); ginekol@ubm.edu.pl (J.S.); 2Department of Medical Biology, Medical University of Bialystok, 15-276 Bialystok, Poland; biollek@umb.edu.pl; 3Department of Endocrinology, Diabetology and Internal Medicine, Medical University of Bialystok, 15-276 Bialystok, Poland; ednodiab@umb.edu.pl; 4Clinical Research Centre, Medical University of Bialystok, 15-276 Bialystok, Poland; 5Department of Hygiene, Epidemiology and Metabolic Disorders, Medical University of Bialystok, 15-276 Bialystok, Poland; higienametz@umb.edu.pl

**Keywords:** gestational diabetes, sphingolipids, ceramides, lipidomic, C18:1-Cer

## Abstract

We hypothesized that sphingolipids may be early biomarkers of gestational diabetes mellitus (GDM). Here, 520 women with normal fasting plasma glucose levels were recruited in the first trimester and tested with a 75 g oral glucose tolerance test in the 24th–28th week of pregnancy. Serum sphingolipids concentrations were measured in the first and the second trimester by ultra-high performance liquid chromatography coupled with triple quadrupole mass spectrometry (UHPLC/MS/MS) in 53 patients who were diagnosed with GDM, as well as 82 pregnant women with normal glucose tolerance (NGT) and 32 non-pregnant women. In the first trimester, pregnant women showed higher concentrations of C16:0, C18:1, C22:0, C24:1, and C24:0-Cer and lower levels of sphinganine (SPA) and sphingosine-1-phosphate (S1P) compared to non-pregnant women. During pregnancy, we observed significant changes in C16:0, C18:0, C18:1, and C24:1-Cer levels in the GDM group and C18:1 and C24:0-Cer in NGT. The GDM (pre-conversion) and NGT groups in the first trimester differed solely in the levels of C18:1-Cer (AUC = 0.702 *p* = 0.008), also considering glycemia. Thus, C18:1-Cer revealed its potential as a GDM biomarker. Sphingolipids are known to be a modulator of insulin resistance, and our results indicate that ceramide measurements in early pregnancy may help with GDM screening.

## 1. Introduction

Gestational diabetes mellitus (GDM) is defined as glucose intolerance first recognized during pregnancy. It is usually the result of β-cell dysfunction on a background of chronic insulin resistance [1]. Other factors affecting insulin sensitivity are maternal obesity, extensive hormones release, adipocytokine production, genetic and epigenetic changes, and novel potential omics factors [2]. Hyperinsulinemic-euglycemic clamp studies in healthy, lean women show that insulin sensitivity is reduced by 56% compared with pre-pregnancy, and basal endogenous glucose production is increased by 30% in the third trimester [3,4]. During pregnancy, we observe an increase in lipid concentration, especially in triglycerides, and, to a lesser extent, phospholipids and cholesterol. It is the result of altered maternal metabolism [5].

Insulin resistance and hyperlipidemia are important physiological processes essential during pregnancy to ensure sufficient fetal nutrition. In women with GDM, the physiological changes in insulin and lipids are excessive but also transitional and may indicate underlying metabolic dysfunction [6,7]. As is well known, lipids perform a crucial role in the biology of the human body, not only being an energy storage and a component of cell membranes, but also acting as an agent in signaling pathways and altering the metabolism. Disturbances in lipid metabolic signaling pathways are associated with inflammation and systemic diseases such as the metabolic syndrome and hypertension [8]. Population-based lipidomic studies indicate that a number of ceramides (Cers), sphingomyelins, and lactosyloceramides are significantly downregulated years before type 2 diabetes onset, suggesting that the downregulation of sphingolipid metabolism could be partially responsible for the future onset of type 2 diabetes among women with GDM history [9]. Furse et al. showed that lipid metabolism was altered at least 10 weeks before a clinical diagnosis of GDM was made [10].

Sphingolipids are a group of biologically active lipids involved in regulation of various cellular processes including cell migration, inflammatory response, proliferation, differentiation, and apoptosis [11,12]. The central molecule in sphingolipid metabolism and the precursor for the complex sphingolipids is ceramide. Available data suggest that it is also a major contributing factor of insulin resistance in skeletal muscles and the liver [13,14,15]. These compounds induce insulin resistance at the level of RACα serine/threonine-protein kinase, also known as Akt or protein kinase B-PKB [16]. This compound activates protein phosphataseA2 (PPA2) and directly catalyzes PKB/Akt dephosphorylation, thus, inhibiting the activity of the insulin pathway [16]. Moreover, type 2 diabetes is often associated with chronic, moderate inflammation. Sphingosine-1-phosphate (S1P) belongs to the sphingolipid family and is a pro-inflammatory compound that increases the expression and secretion of cytokines (e.g., TNFα, IL-6, MCP-1) [17]. However, the effect of S1P on the inflammatory response has been demonstrated to be dependent on a carrier protein. The major carrier proteins for S1P are apolipoprotein M (apoM) and albumin. Most of the plasma S1P is bound to the apoM/ApoM-S1P that binds preferentially to HDL. ApoM-S1P has been shown to inhibit inflammatory responses in endothelial cells [18]. These features suggest that S1P may induce the disorders leading to GDM, but there is little literature data on this subject.

Here, we hypothesized that circulating sphingolipids may be early biomarkers of GDM development. To test this hypothesis, serum sphingolipid levels were measured in the first and the second trimester and compared between the patients with normal glucose tolerance (NGT) and GDM diagnosed between 24 and 28 weeks of pregnancy.

## 2. Materials and Methods

### 2.1. Study Population

Women (n = 520) with normal fasting plasma glucose levels (<92 mg/dL (5.1 mmol/L)) were recruited in the first trimester of pregnancy from the Gynecological Out-Patient Clinic of the Medical University of Bialystok. Women with a history of GDM, stillbirth, congenital anomalies, pregnancy-induced hypertension, multiple pregnancy, pre-existing glucose intolerance, or acute or chronic inflammation and active smokers were not included. All patients underwent a 75 g oral glucose tolerance test (OGTT) in the 24th–28th week of pregnancy and gestational diabetes was diagnosed in 53 women (GDM) according to the WHO 2013 criteria [19]. Their results were compared with the results of the carefully selected 82 pregnant women with normal glucose tolerance (NGT). We also enrolled a third control group that consisted of 37 healthy, non-pregnant women. All groups were matched for age, and pre-pregnancy BMI was calculated as weight in kilograms divided by height in meters. Written informed consent was obtained from all participants before enrolment, and the protocol was approved by the local ethics committee (Medical University of Bialystok).

### 2.2. Diabetic Parameters

In the 1st trimester, venous blood samples were collected in the fasting state. Serum was collected by allowing freshly drawn blood to clot, followed by centrifugation at 2000× *g* for 10 min in a refrigerated centrifuge. The resulting supernatant was collected and stored at −80 °C until further analysis. The 75 g oral glucose tolerance test (OGTT) was performed in the 24th–28th week of pregnancy in the pregnant patients, as well as in the control group, after an overnight fast. Blood samples were collected at 0, 30, 60, and 120 min after glucose load. Plasma glucose concentration was measured via an enzymatic method with hexokinase (Cobas c111, Roche Diagnostics Ltd., Switzerland). Serum insulin levels were assayed by immunoradiometric method (DiaSource Europe SA, Belgium), and glycated hemoglobin (HbA1c) was evaluated with a high-performance liquid chromatography technique (BIO-RAD Laboratories, Germany). The following indices of insulin sensitivity and insulin secretion were calculated: (1) HOMA-IR (the homeostasis model assessment of insulin resistance) = FPG (mmol/L) × FPI (µU/mL)/22.5, HOMA-β [%] = 20 × FPI (mU/L)/FPG (mmol/L) − 3.5; and (2) the Matsuda and de Fronzo index (ISOGTT) = 10,000/√ ((FPG × FPI) × (G × I)), where FPG = fasting plasma glucose, FPI = fasting plasma insulin, G = mean glucose, and I = mean insulin during the OGTT [20]. Total cholesterol, HDL-cholesterol, and triglyceride concentrations were measured by enzymatic methods (Cobas c111, Roche Diagnostics Ltd., Rotkreuz, Switzerland). LDL-cholesterol concentration was calculated using the Friedewald equation.

### 2.3. Sphingolipid Measurements

The concentration of serum sphingolipids was measured in the first and the second trimester of pregnancy using a UHPLC/MS/MS approach according to Błachnio-Zabielska et al. [21]. Briefly, an internal standard mix (17C-sphingosine, 17C-S1P, d17:1/8:0, d17:1/18:0, d17:1/18:1 (9Z), d17:1/20:0, d17:1/24:0, and d17:1/24:1 (15Z)) (Avanti Polar Lipids, Alabaster, Al), as well as an extraction mixture (isopropanol:water:ethyl acetate, 30:10:60; v:v:v), was added to each serum sample. The samples were vortexed and then centrifuged. The supernatants were transferred to new tubes and pellets were re-extracted. After centrifugation, supernatants were combined and evaporated under nitrogen. The dried samples were reconstituted in LC Solvent B (2 mM ammonium formate, 0.15% formic acid in methanol) for UHPLC/MS/MS analysis (Sciex 6500+; AB Sciex Germany GmbH, Darmstadt, Germany). The chromatographic separation was performed on a reverse-phase Zorbax SB-C8 column 2.1 × 150 mm, 1.8 μm (Agilent Technologies, Santa Clara, CA, USA) in a binary gradient using 1 mM ammonium formate with 0.1% formic acid in water as solvent A and 2 mM ammonium formate and 0.1% formic acid in methanol as solvent B at the flow rate of 0.4 mL/min. Sphingolipids concentrations were analyzed via a triple quadrupole mass spectrometer using positive ion electrospray ionization (ESI) (except for S1P, which was analyzed in negative mode) with multiple reaction monitoring (MRM) against the concentration standard curves.

### 2.4. Statistical Analysis

The data were analyzed by the STATISTICA 13 for Windows (StatSoft. Inc., Tulsa, OK, USA). Data were shown as medians and interquartiles. Differences between the two groups were compared by Mann–Whitney U test, differences between all three groups were compared via a Kruskal–Wallis test, and relationships between variables were tested by Spearman’s correlations. A Wilcoxon signed rank test was used to compare sphingolipids levels in the 1st and 2nd trimester. A *p*-value less than 0.05 was considered statistically significant. Data are shown as medians with interquartile ranges.

## 3. Results

### 3.1. Characteristics of the Studied Groups

Table 1 and Table 2 compare the clinical and biochemical characteristics of the study patients.

In the first trimester, the patients who were later diagnosed with GDM (converters) had no significant differences compared with the women with NGT (Table 1). The comparison of the three groups revealed significant differences between fasting glucose levels (*p* < 0.001), which is a consequence of a variation in norms (19). Fasting insulin (*p* < 0.001), HOMA-IR (*p* < 0.001), HOMA-β (*p* < 0.001), and triglycerides levels (*p* < 0.001) were higher in both pregnant women groups; HDL-cholesterol (*p* = 0.01) was higher in the control group.

In the second trimester, the GDM converters had significantly higher fasting and post-load glucose concentration (*p* < 0.001); higher fasting and post-load insulin concentrations (*p* < 0.001) except for insulin 30′, as well as higher HOMA-IR (*p* < 0.001) and HbA1c (*p* = 0.006). Furthermore, GDM converters had lower ISI OGTT (*p* < 0.001) and lower total cholesterol levels (*p* = 0.04) versus the NGT group (Table 2). 

### 3.2. Sphingolipids Profile

The concentration of total serum ceramides was significantly lower in non-pregnant women versus pregnant ones (*p* = 0.0006 versus the GDM pre-conversion and *p* < 0.0001 versus the NGT group). The control group had lower levels of C16:0-Cer (*p* = 0.02 vs. GDM pre-conversion and *p* = 0.0002 vs. NGT), C18:1-Cer (*p* < 0.0001 vs. GDM pre-conversion and *p* = 0.006 vs. NGT), C22:0-Cer (*p* < 0.0001 in both comparisons), C24:1-Cer (*p* = 0.0001 vs. GDM pre-conversion and *p* < 0.0001 vs. NGT), and C24:0-Cer (*p* = 0.03 vs. GDM pre-conversion and *p* = 0.003 vs. NGT). The control group had higher sphinganine (SPA) and S1P levels (*p* < 0.0001 in both comparisons vs. GDM pre-conversion and the NGT group). 

Among the pregnant women, comparisons between the GDM converter group and the NGT group using the Mann–Whitney U test revealed prominent differences in C18:1 concentration (*p* = 0.01) (Table 3, Figure 1). The diagnostic value of this ceramides species was evaluated by ROC analysis. The values of AUC and optimal cut-off value for C18:1-Cer were as follows: 0.702 confidence interval: 0.552–0.852, *p* = 0.008 (Figure 2).

Across the study population, the SPA concentration correlated negatively with serum insulin (R = −0.33, *p* < 0.05), HOMA-IR (R = −0.3, *p* < 0.05), and HOMA-β (R = −0.38, *p* < 0.05). Positive correlations with serum insulin parameters were observed in C22:0-Cer with insulin (R = 0.3, *p* < 0.05), HOMA-IR (R = 0.3, *p* < 0.05), and HOMA-β (R = 0.31, *p* < 0.05); C24:0-Cer correlated with insulin (R = 0.27, *p* < 0.05), HOMA-IR (R = 0.23, *p* < 0.05), and HOMA-β (R = 0.32, *p* < 0.05). Furthermore, there were correlations between C18:1-Cer and serum insulin levels (R = 0.21, *p* ≤ 0.05) and HOMA-β (R = 0.21, *p* ≤ 0.05), as well as between C24:1-Cer and insulin level (R = 0.22, *p* ≤ 0.05) and HOMA-β (R = 0.29, *p* ≤ 0.05). 

Wilcoxon analysis demonstrated progressive changes in ceramide concentration levels during pregnancy (Table 4). We compared measurements from the first trimester with measurements from the second trimester. The most relevant change was observed in NGT C24:0-Cer in the second trimester. This appeared to be higher compared to the first (*p* < 0.001). Levels of C16:0-Cer and C18:0-Cer in the GDM group were increased while these parameters were stable in the NGT group. C24:1-Cer in the GDM group was higher in the first trimester than in the second; the NGT group had the opposite situation, but it was not significant. Moreover, the concentration of C18:1-Cer in GDM decreased during pregnancy, but it increased in NGT.

Comparison of sphingolipid concentrations measured in the second trimester of pregnancy showed prominent differences between the GDM and NGT group in terms of S1P (*p* = 0.009), C16:0-Cer (*p* = 0.04), C18:0-Cer (*p* = 0.0002), and C24:1-Cer (*p* = 0.03). The diagnostic value of those sphingolipids was evaluated by ROC analysis. The values of AUC and optimal cut-off value for S1P were 0.638 (95% confidence interval: 0.541–0.735; *p* = 0.005) and 304.15 ng/mL. AUC was 0.606 (95% confidence interval: 0.508–0.705; *p* = 0.03) and the cut-off value was 184.67 ng/mL for C16:0-Cer. AUC was 0.696 (95% confidence interval: 0.6–0.792; *p* = 0.0001) and the cut-off value was 155.95 ng/mL for C18:0-Cer. AUC = 0.618 (95% confidence interval: 0.519–0.718; *p* = 0.02) and the cut-off value was 272.16 ng/mL for C24:1-Cer.

## 4. Discussion

The data show that the main pathophysiological dysfunction in GDM is the increasing insulin resistance and insufficient insulin compensation, usually as a result of the β-cell impairment [1,22]. Sphingolipids, particularly ceramides, are potential factors contributing to diabetes [23]. Our study was designed to measure and compare the changes in circulating sphingolipids concentration in pregnancy and in GDM. It is worth mentioning that, according to O’Sullivan, Carpenter and Coustan [24] and some associations, such as the Spanish Group of Diabetes and Pregnancy (GEDE) [25], OGTT should be taken with 100 g of glucose and measured in a fasting state, 60, 120, and 180 min after overload. However, Cabrera-Fernandez et al. [26] revealed that using the GEDE criteria measurement at 180 min could be omitted. In our work, we applied criteria recommended by the International Association of Diabetes and Pregnancy Study Group (IADPSG) [24] and the International Diabetes Federation (IDF) [27], where 75 g of glucose is used, and the test lasts 120 min.

This study showed that the total concentration of circulating ceramides measured in the first trimester of pregnancy was significantly higher in pregnant patients versus healthy, non-pregnant women. The pregnant women group showed a significant increase in C16:0-Cer, C 18:1-Cer, C22:0-Cer, C24:1-Cer, and C24:0-Cer. It is known that ceramide biosynthesis is enhanced by insulin in skeletal muscles [28]. During pregnancy, insulin resistance increases, and the same insulin concentration is higher. This might explain the increasing ceramide levels.

Elevated C16:0-Cer concentrations have been found in overweight patients and those with type 2 diabetes [29,30]. Here, we excluded the possibility of obesity influencing the results. The groups were selected so that they did not present differences in the BMI. However, we show significantly increased C16:0-Cer levels between the first trimester and the second trimester in patients who developed GDM. Raichur et al. [30] reported that the inhibition of synthesis of C16:0-Cer improved insulin resistance and decreased hyperglycemia. They also reported that C16:0-Cer is an important factor in diabetes development, and our study confirmed this theory.

Available data report an association between an increased risk of diabetes development and higher C18:0-Cer levels in plasma [31]. Similarly, in our experiment, in the GDM group this parameter increased during pregnancy and, in the second trimester, was revealed to be significantly higher than in the NGT group where it stayed at a constant level. 

Other authors reported increased C18:0, C18:1, C20:0, C22:0, and C24:0-Cer in subjects with a glucose tolerance impairment phenotype [10,32]. They also reported a positive association between circulating ceramide levels and insulin-resistance parameters including disruption of β-cell function and inflammation [32]. During pregnancy, even physiological pregnancy, there is an increase in insulin resistance and a prothrombotic state. This showed pro-inflammatory features [33]. We confirmed a positive correlation between C22:0-Cer and insulin levels and HOMA-IR, as well as between C18:1, C24:0, and C24:1 with insulin concentrations and HOMA β across the entire study population. Considering the subgroups, we noted that the GDM converters group’s C24:1 positively correlated with insulin concentration and HOMA β.

Khan et al. [9] recently employed artificial intelligence and demonstrated the predictive value of reduced ceramide levels in the pathophysiology of transition from GDM to type 2 diabetes. Suppression of ceramide synthesis in pancreatic islets impairs glucose-stimulated insulin secretion. This discovery suggests that any imbalance may contribute to insulin resistance. Further research into the role of ceramides and mechanisms underlying GDM and diabetes development—especially in the pathophysiology of β-cells dysfunction and insulin resistance—is required.

Early in pregnancy, fat accumulation occurs due to an increased lipid synthesis and hyperphagia; in the last trimester of pregnancy, fat accumulation halts due to decreased adipose tissue lipoprotein lipase activity [5]. Adipose tissue is an endocrine organ, and its adipocytes synthesize i.a. adiponectin. This adipokine was found to elicit broad spectrum antidiabetic action by activating ceramidase to degrade ceramides [34]. Adiponectin receptors have a homology with ceramidase enzymes. They can activate or deplete adiponectin receptors to markedly alter cellular ceramidase activity [35]. It is known that decreased levels of adiponectin during pregnancy indicates an increased risk of GDM [36]. 

The pregnant women in our study had lower SPA and S1P levels compared to healthy, non-pregnant women. It is known that S1P is a powerful bioactive lipid that can act intracellularly and extracellularly, and its receptors are widely expressed in the human body [37]. S1P is carried on apolipoprotein M (apoM) [8], a minor apolipoprotein on HDL [9]; it has been proposed to be responsible for many of the pleiotropic qualities of HDL, i.e., having antiapoptotic [10], anti-inflammatory [38], and vasoprotective effects [39]. 

Recent studies demonstrated the crucial role of S1P in insulin sensitivity and glucose homeostasis. Kurano et al. [40] provided evidence that at least a part of HDL’s antidiabetic action involves apolipoprotein M (apoM) and its lipid ligand sphingosine-1-phosphate (S1P); these are two quantitatively minor components of HDL. They demonstrated that apoM/S1P protects against the development of insulin resistance in the liver, adipose tissue, and skeletal muscles by activating AKT and AMPK signaling, which are the main signaling pathways and act via S1P1 and/or S1P3 by enhancing mitochondrial functions, perhaps through the upregulation of the SIRT1 protein levels. 

A prior study with an animal model showed that increased glycemia resulted in activation of sphingosine kinase isoform 2 (Sphk2) in pancreatic β-cells and prominently increased S1P. Knockdown of Sphk2 led to the abolition of insulin secretion in response to glucose [41]. Liu et al. [42] proved that S1P prevents β-cell apoptosis—thus, the authors suggested inhibition of the voltage-dependent potassium channel in pancreatic β-cells, which also induces Ca^2+^ inflow into the cell to stimulate insulin secretion.

The literature suggests elevated levels of SPA in type 2 diabetes patients versus healthy control subjects [43]. Moreover, SPA was shown to negatively correlate with insulin sensitivity [44]. In contrast to those reports, we present, here, decreased levels of SPA in the GDM group versus the control group. This may be due to the use of SPA for ceramide synthesis—it is a major precursor in the de novo synthesis pathway [45].

Interestingly, our study demonstrated that the C18:1-Cer level was significantly higher in GDM converters than in the NGT group in the first trimester. It is worth emphasizing that this was the only difference between those groups considering measured sphingolipids, as well as clinical/biochemical parameters. Thus, we suspect that C18:1-Cer may be a potential GDM biomarker. It can manifest a predisposition for disease development before glycemic changes occur. Our results demonstrate that ceramide levels are elevated in pregnancy and changes in the ceramide profile are not independent of BMI. We are aware of only one report about circulating ceramides in early pregnancy. Liu et al. [46] selected the studied groups (n = 486) from a large cohort of Chinese women (n = 22302). The differences in concentration of ceramides in early pregnancy are significant. Scientists noted that three ceramides were significantly associated with GDM development in later pregnancy. The GDM group showed an increased level of C18:0-Cer and C18:1-Cer and a decreased level of C24:0-Cer. Our results partially confirm this data. We proved only an increase of C18:1-Cer—the remining ceramides showed no significant differences in our groups. Further analysis of ceramides in the second trimester demonstrated that C18:1-Cer concentrations decreased in the GDM serum, but this increased in the NGT group. In contrast to Liu et al., we studied smaller but BMI-matched groups of varying ethnicities. 

In summary, our results show differences in the metabolic signature between GDM and control pregnant group in the second trimester of pregnancy. The results emphasize the potential of C 18:1-Cer in the first trimester of pregnancy as a lipidomic biomarker of GDM development. The main limitation of this research was the low number of patients. Furthermore, we did not control for patient diet, which can impact circulating ceramides [47]. Further studies are required to validate our data and to clarify and improve the understanding of GDM pathophysiology.

## Figures and Tables

**Figure 1 jcm-11-00384-f001:**
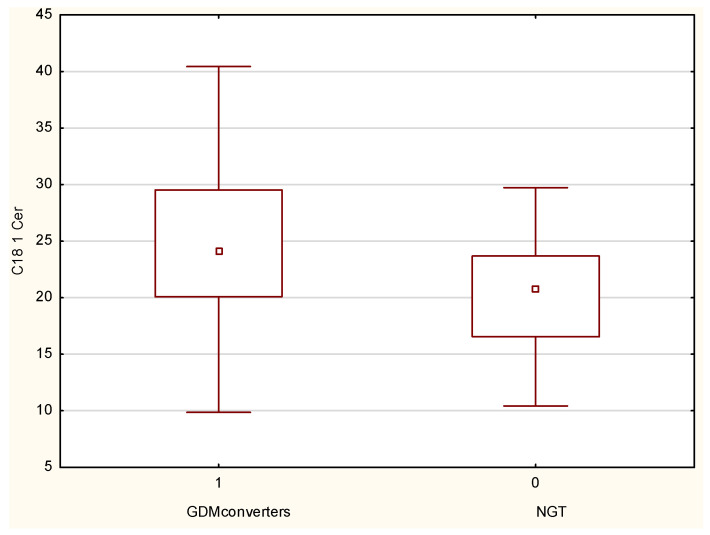
Serum C18:1-Cer concentrations measured in the first trimester of pregnancy. Data are shown as medians and interquartile range.

**Figure 2 jcm-11-00384-f002:**
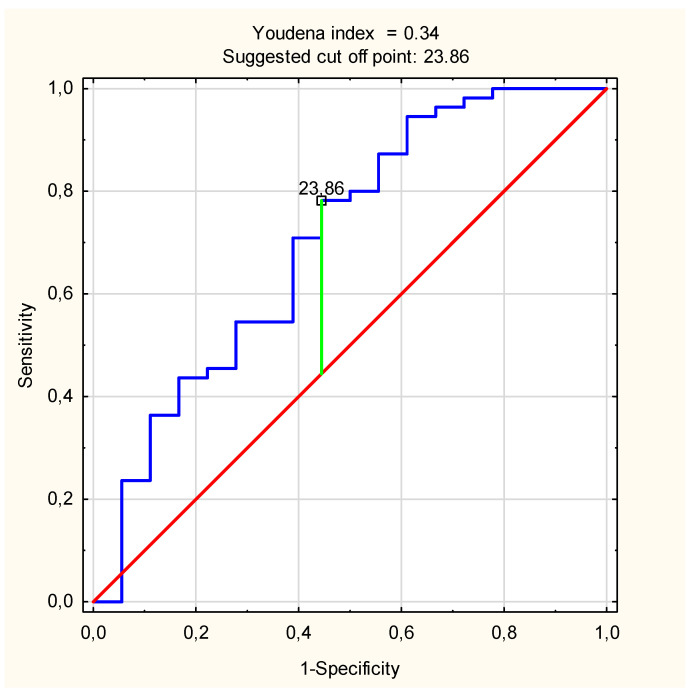
ROC curve for C18:1-Cer measured in the 1st trimester of pregnancy (AUC = 0.702 (95% confidence interval: 0.552–0.852; *p* = 0.008) with optimal cut-off value of 23.87 ng/mL).

**Table 1 jcm-11-00384-t001:** Clinical characteristics of the groups studied in the 1st trimester.

	Control	NGT	GDM (Pre-Conversion)	*p*-Value
*n*	37	82	53	
Age (years)	26 (23–31)	28 (24–32)	25.5 (24–30)	0.4 ° 0.81 *
Gestational age (week)		11 (10–12)	11 (10–11)	0.19 *
Prepregnancy BMI (kg/m^2^)	21.9 (20.6–23.4)	20.9 (19.8–28.5)	24.1 (21.6–26.8)	0.42 ° 0.87 *
Current BMI (kg/m^2^)		24.5 (20.4–29.8)	24.8 (22.0–26.7)	0.95 *
Fasting glucose (mg/dL)	90 (86–92)	86 (84–88)	87 (84.5–89.5)	0.0005 ° 0.19 *
Fasting insulin (µU/mL)	7.5 (5.2–10.7)	11.6 (8.9–14.7)	11.3 (10.1–13.3)	<0.0001 ° 0.83 *
HOMA–IR	1.6 (1.1–2.4)	2.4 (1.8–3.2)	2.5 (2.1–2.9)	0.0002 ° 0.89 *
HOMA–β	101.1 (67.3–149)	176.9 (151.9–226.8)	173.7 (155.7–220.0)	<0.0001 ° 0.82 *
HbA1c (%)	5.2 (5.0–5.4)	5.0 (4.9–5.3)	5.1 (4.9–5.4)	0.2 ° 0.31 *
Total cholesterol (mmol/L)	166 (158–182)	174 (150–202)	172 (156.5–187)	0.64 ° 0.64 *
HDL-cholesterol (mmol/L)	86 (69.6–102.8)	73 (63–88)	72.5 (59.5–80.5)	0.01 ° 0.49 *
LDL-cholesterol (mmol/L)	64 (53–83)	78 (63.4–96.6)	87.6 (69.3–95.5)	0.03 ° 0.79 *
Triglycerides (mmol/L)	69 (59–75)	87 (65–116)	84.5 (64.5–125.5)	0.0009 ° 0.92 *

Data are shown as medians (interquartile range); ° differences between all groups were analyzed by Kruskal–Wallis test. * The difference between NGT and GDM groups was compared by Mann–Whitney U test.

**Table 2 jcm-11-00384-t002:** Clinical characteristics of the groups studied in the 2nd trimester.

	NGT	GDM	*p*–Value
*n*	82	53	
Gestational age (week)	25 (24–26)	25.5 (24–26)	0.41
Current BMI (kg/m^2^)	26.17 (22.6–31.8)	27.8 (23.7–29.7)	0.76
Fasting glucose (mg/dL)	83 (80–86)	94 (89–97)	<0.0001
Glucose 30’ (mg/dL)	127 (117–139)	157.5 (139.5–166)	<0.0001
Glucose 60’ (mg/dL)	121.5 (103–139)	164 (129.5–184)	<0.0001
Glucose 120’ (mg/dL)	108 (89–121)	124 (113–157)	<0.0001
Fasting insulin (µU/mL)	11.1 (9.2–14.2)	15.9 (12.8–20.5)	<0.0001
Insulin 30’ (µU/mL)	77.9 (56.2–115.1)	89.3 (61.1–137.3)	0.18
Insulin 60’ (µU/mL)	72.3 (47.9–112.8)	108.3 (91.1–152.4)	<0.0001
Insulin 120’ (µU/mL)	56.0 (35.2–78.0)	104.8 (68.7–131)	<0.0001
HOMA–IR	2.3 (1.9–2.9)	3.6 (2.8–4.7)	<0.0001
HOMA–β	223.2 (169.8–276.1)	201.4 (154.4–244.3)	0.13
ISI OGTT	4.3 (3.3–5.4)	2.6 (2.1–3.4)	<0.0001
HbA1c (%)	4.8 (4.7–5.1)	5.0 (4.8–5.2)	0.006
Total cholesterol (mmol/L)	243 (219–276)	233 (203–257)	0.04
HDL-cholesterol (mmol/L)	86 (73–95)	79 (65–93)	0.13
LDL-cholesterol (mmol/L)	120.4 (95.6–149.2)	119.2 (86.8–134.4)	0.08
Triglycerides (mmol/L)	172 (133–196)	179 (136–219)	0.31

Data are shown as medians (interquartile range); the difference between NGT and GDM groups was compared the Mann–Whitney U test.

**Table 3 jcm-11-00384-t003:** The concentration of sphingolipids in the serum of patients in the non-pregnant control group and pregnant GDM pre-conversion and NGT group in their 1st trimester.

Compound	Control Group	NGT	GDM Pre-Conversion	*p*-Value
	Me (Q1–Q3) [ng/mL]	Me (Q1–Q3) [ng/mL]	Me (Q1–Q3) [ng/mL]	
Sph	23.4 (20.6–26.4)	18.02 (13.3–31.7)	17.2 (14.5–36.9)	* *p* = 0.73 ° *p* = 0.2 ᶺ *p* = 0.09
SPA	37.5 (34.2–43.8)	13.89 (9.9–19.7)	10.82 (8.0–20.2)	* *p* = 0.28 ° *p* < 0.0001 ᶺ *p* < 0.0001
S1P	400.93 (357.2–436.8)	348.81 (251.4–403.4)	296.78 (235.3–342.4)	* *p* = 0.06 ° *p* < 0.0001 ᶺ *p* < 0.0001
C14:0 Cer	30.56 (26.5–34.7)	32.4 (25.6–42.9)	29.6 (23.4–36.7)	* *p* = 0.22 ° *p* = 0.5 ᶺ *p* = 0.19
C16:0 Cer	135.68 (118.6–156.6)	183.37 (151.3–246.6)	178.01 (129.9–201.4)	* *p* = 0.49 ° *p* = 0.02 ᶺ *p* = 0.0002
C18:1 Cer	16.33 (14.3–18.5)	20.72 (16.6–23.7)	24.16 (20.1–29.5)	* *p* = 0.01 ° *p* < 0.0001 ᶺ *p* = 0.006
C18:0 Cer	127.75 (114.3–142.2)	133.01 (108.4–160.5)	132.87 (108.1–199.6)	* *p* = 0.57 ° *p* = 0.6 ᶺ *p* = 0.5
C20:0 Cer	183.56 (156.4–206.7)	172.96 (130.1–194.2)	152.08 (123.6–177.9)	* *p* = 0.25 ° *p* = 0.09 ᶺ *p* = 0.3
C22:0 Cer	215.53 (198.3–240.4)	280.68 (243.5–317.7)	316.04 (256.0–376.7)	* *p* = 0.1 ° *p* < 0.0001 ᶺ *p* < 0.0001
C24:1 Cer	219.88 (200.8–251.5)	280.36 (237.9–336.5)	278.96 (241.5–320.3)	* *p* = 0.92 ° *p* = 0.0001 ᶺ *p* < 0.0001
C24:0 Cer	1941.61 (1819.1–2306.9)	2200.14 (2040.8–2608.7)	2356.23 (2026.8–2917.9)	* *p* = 0.52 ° *p* = 0.03 ᶺ *p* = 0.003
Cer Total	3023.67 (2738.7–3225.9)	3344.21 (3101.9–3682.7)	3552.17 (3007.1–3923.5)	* *p* = 0.39 ° *p* = 0.0006 ᶺ *p* < 0.0001

Data are presented as medians (interquartile range). Analysis was performed with the Mann–Whitney U test. * GDM vs. NGT; ° GDM vs. control; ᶺ NGT vs. control.

**Table 4 jcm-11-00384-t004:** The concentration of sphingolipids in the serum of patients from the GDM and NGT groups by trimester.

Compound	NGT				GDM Converters			
	1st Trimester		2nd Trimester		1st Trimester		2nd Trimester	
	Me	Q1–Q3	Me	Q1–Q3	Me	Q1–Q3	Me	Q1–Q3
Sph	18.02	13.3–31.7	18.7	15.3–23.7	17.2	14.5–36.9	18.07	15.4–20.8
SPA	13.89	9.9–19.7	16.14	12.6–21.5	10.82	8.0–20.2	14.71	10.9–18.5
S1P	348.81	251.4–403.4	307.53	214.1–423.3	296.78	235.3–342.4	263.73	180.9–304.2
C14:0 Cer	32.4	25.6–42.9	35.87	23.9–42.6	29.6	23.4–36.7	35.47	27.8–43.4
C16:0 Cer	183.37	151.3–246.6	184.34	133.8–234.7	178.01 *	129.9–201.4	214.38 *	178.5–250.2
C18:1 Cer	20.72 *	16.6–23.7	22.36 *	17.7–25.9	24.16 *	20.1–29.5	20.36 *	15.6–26.5
C18:0 Cer	133.01	108.4–160.5	137.56	116.9–154.6	132.87 *	108.8–199.6	168.21 *	138.4–201.0
C20:0 Cer	172.96	130.1–194.2	174.51	136.7–210.6	152.08	123.6–177.9	180.41	146.8–242.8
C22:0 Cer	280.68	243.5–317.7	296.23	248.7–347.2	316.04	256.0–376.7	278.25	237.0–336.5
C24:1 Cer	280.36	237.9–336.5	285.22	234.5–344.5	278.96 *	241.5–320.3	257.62 *	214.8–296.6
C24:0 Cer	2200.14 °	2040.8–2608.7	2612.07 °	2306.4–2936.5	2356.23	2026.8–2917.9	2545.29	2115.7–3077.4
Cer Total	3344.21	3101.9–3682.7	3731.5	3494.8–4298.8	3552.17	3007.1–3923.5	3737.46	3310.1–4454.1

This table shows changes during pregnancy. Data are presented as medians and interquartile range; * *p* < 0.05; ° *p* < 0.001. Analysis was performed with Wilcoxon test.

## Data Availability

The datasets used and analyzed during the current study are available from the corresponding author on reasonable request.

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
