# Peer review of "Serum C18:1-Cer as a Potential Biomarker for Early Detection of Gestational Diabetes"

_jcm, 2022, doi:10.3390/jcm11020384_

Round 1

Reviewer 1 Report

It is an interesting article. But I would like to make a few comments.
Strict and clear introduction. These researchers, authors of a very good review, should be cited.

Mirghani Dirar A, Doupis J. Gestational diabetes from A to Z. World J Diabetes. 2017 Dec 15;8(12):489-511. doi: 10.4239/wjd.v8.i12.489. PMID: 29290922; PMCID: PMC5740094.

Methods: In the methodology, it seems to be a very good option that the comparisons have been made with non-parametric methods (Mann-Whitney U test and Kruskal-Wallis test) to avoid the problems of a relatively small sample size. 

What I no longer understand is that in the multivariate analysis purely parametric methods such as linear regression have been used without making explicit that they fulfill the necessary prerogatives for this form of study.  Perhaps binary logistic regression was a good solution to that problem because the usage requirements are less rigid.
The response variable would be to suffer or not have gestational diabetes.

Another important issue is the level of glucose overload. The authors state 75 mg when the normal clinical overload is 100 mg, and the measurements are in the basal state, at 60, 120, and 180 minutes. The action taken by the authors is correct but they should expose the differences.
This work should be cited in that place:

Cabrera Fernández S, Martín Martínez MD, De Francisco Montero C, Gabaldón Rodríguez I, Vilches Arenas Á, Ortega Calvo M : Predictive models of gestational diabetes, a new prediction mode.  Semergen. 2021 Nov-Dec;47(8):515-520. Spanish. doi: 10.1016/j.semerg.2021.07.014. PMID: 34509372.

Author Response

The authors would like to thank the Reviewer for the careful evaluation and all the valuable remarks which helped us to improve the manuscript.

In response to the comments, we would like to thank you for your kind appreciation for choosing non-parametric methods for group comparison. After consulting with our statistician, we decided to remove the linear regression models from our manuscript, thank you for paying attention to an such important issue.

The difference in various diagnostic methods and diagnostic criteria for gestational diabetes mellitus has been added in the first paragraph of the discussion section. Two suggested articles have been citated.

The changes within the manuscript are highlighted by using bold.

Reviewer 2 Report

The authors emphasize the important role of Serum C18.1 ceramide as a potential maker for the detecting of early GDM. However, the study raises a major concern about the study groups.

The OGTT performed in the 2nd trimester very much correlates to the difference in BMI NGT 20.9 and GDM 24.1. In order to rule out the possibilities of BMI interference, the authors need to replot a supplement table dividing the population with comparable BMI and reanalyze the parameters to show that there is a real difference. A 5% difference in BMI has been show to make a clear difference in many insulin related parameters.  

Author Response

The authors are grateful for the careful evaluation of our manuscript by the Reviewer and for the crucial comments, which helped us to improve the manuscript.

            Dividing the GDM converters into smaller categories in terms of BMI rages would result in a formation of subgroups with a small number of subjects, thus affecting the statistical analysis. Both NGT and GDM converters were matched for age and pre-pregnancy BMI with medians in the range 18.5 to 24.9 which is considered as a normal range. Additionally, we added to the Table 1 and Table 2 a current BMI value in the first and second trimester of pregnancy. Both groups did not differ in terms of current BMI (in the first trimester of pregnancy: NGT: Me=24.5; GDM: Me=24.8; p-value=0.95; in the second trimester of pregnancy: NGT: Me=26.17; GDM: Me=27.8; p-value=0.76).

The changes within the manuscript are highlighted by using bold.

Round 2

Reviewer 1 Report

line 253: ... However , Cabrera-Fernández et al (26) , revealed ...

Reviewer 2 Report

NA

This manuscript is a resubmission of an earlier submission. The following is a list of the peer review reports and author responses from that submission.